Harvest year effects on Apulian EVOOs evaluated by 1H NMR based metabolomics

Girelli Chiara R.
Del Coco Laura
Papadia Paride paride.papadia@unisalento.it
De Pascali Sandra A.
Fanizzi Francesco P. fp.fanizzi@unisalento.it
University of Salento, Dipartimento di Scienze e Tecnologie Biologiche e Ambientali (Di.S.Te.B.A.) , Lecce , Italy
Benelli Giovanni
Electronic publication date: 2016 Dec 15
Publication date: 2016
Volume: 4
Electronic Location ID: e2740
Received 2016 Aug 19; Accepted 2016 Oct 25
Copyright: ©2016 Girelli et al.
Copyright year: 2016
Copyright holder: Girelli et al.
License: This is an open access article distributed under the terms of the Creative Commons Attribution License, which permits unrestricted use, distribution, reproduction and adaptation in any medium and for any purpose provided that it is properly attributed. For attribution, the original author(s), title, publication source (PeerJ) and either DOI or URL of the article must be cited.
License URL: https://creativecommons.org/licenses/by/4.0/

Keywords: Nuclear Magnetic Resonance (NMR), Harvest year effect, Triacilglycerols (TAG), Extra-virgin olive oil (EVOO), Cluster difference, Single-cultivar, Mahalanobis distance (MAH), Quality metrics

Funding: Italian Ministry of University and Research (MIUR) PON01_01958 PIVOLIO This study was supported by the Italian Ministry of University and Research (MIUR), Project PON “R&C” 2007–2013 (PON01_01958 PIVOLIO). The funders had no role in study design, data collection and analysis, decision to publish, or preparation of the manuscript.

==============================
Nine hundred extra virgin olive oils (EVOO) were extracted from individual olive trees of four olive cultivars (Coratina, Cima di Mola, Ogliarola, Peranzana), originating from the provinces of Bari and Foggia (Apulia region, Southern Italy) and collected during two consecutive harvesting seasons (2013/14 and 2014/15). Following genetic identification of individual olive trees, a detailed Apulian EVOO NMR database was built using 900 oils samples obtained from 900 cultivar certified single trees. A study on the olive oil lipid profile was carried out by statistical multivariate analysis (Principal Component Analysis, PCA, Partial Least-Squares Discriminant Analysis, PLS-DA, Orthogonal Partial Least-Squares Discriminant Analysis, OPLS-DA). Influence of cultivar and weather conditions, such as the summer rainfall, on the oil metabolic profile have been evaluated. Mahalanobis distances and J2 criterion have been measured to assess the quality of resulting scores clusters for each cultivar in the two harvesting campaigns. The four studied cultivars showed non homogeneous behavior. Notwithstanding the geographical spread and the wide number of samples, Coratina showed a consistent behavior of its metabolic profile in the two considered harvests. Among the other three Peranzana showed the second more consistent behavior, while Cima di Mola and Ogliarola having the biggest change over the two years.

Introduction

Olive trees were originally (thousands of years ago) found only in the Mediterranean region, while today are grown in several countries around the world. In addition to its primeval European and African Mediterranean countries, nowadays other production areas are Argentina, South Africa and Australia (http://www.madeinsouthitalytoday.com, 2016). Nevertheless, still about 95% of total olive oil is produced in the Mediterranean region (Europe and North Africa). In the Mediterranean area Italy can be considered, for its environmental, historical conditions, and high variety of cultivars, a key country for the olive oil production. Average olive oil production in the EU in recent years was estimated in 2.2 million tonnes, representing around 73% of world production. Spain, Italy and Greece account for about 97% of EU olive oil production, with Spain producing approximately 62% of this amount (European Commission, Agriculture and Rural Development, 2016). The weather conditions heavily affected the European olive production in the 2014/15 harvest season. In particular, European Union olive oil production collapsed in 2014, with a −17% as compared to 2013. According to Ismea, the Italian institute for agricultural and food market services, and AIFO, CNO, UNASCO and UNAPROL, a group of trade organizations, the Italian production of olive oil has fallen 35% in the 2014/15 season, to 302,000 tonnes from 463,000 of the previous year (ISMEA, 2016). The effects of this decline in production have been the dramatic rise in prices and the increased risks of scams and frauds. Since 60.4% of Italian olive oil comes from Apulia Region (this value reaches 90% of the entire oil production when considering together also Sicily and Calabria), there is a real need to define the characteristics of Apulian olive oil production. In particular for Apulia region (Southeast Italy), the production of olive oil shows a strong relation with the used cultivars and local pedoclimatic conditions. The decreased production in the 2014/15 (H14) with respect to in the 2013/14 (H13) harvest season was the result of adverse weather conditions, such as high humidity and low summer temperatures that also facilitated attacks by pathogens. Annual precipitation (cumulative rainfall) and temperature were different for the two harvest years, especially over the summer months with the 2014/2015 having high rainfall (for quick reference, we summarized weather data collected from ISPRA (2016), Higher Institute for Environmental Protection and Research, in Fig. S1). In addition, the infestation of the fly Bactrocera oleae was favoured by the anomalous climatic conditions in 2014/15 year such as a very wet and very fresh summer followed by an exceptionally mild autumn and winter (the warmest in the last 60 years) with no frosts. In the 2014 summer, the relatively cool climate has allowed the fly to stay active even during the typically hottest months, July and August, when it is normally inhibited. At the same time the high rainfall has sometimes decreased the effectiveness of treatments (due to washout) and prevented the opportune field interventions due to their inaccessibility. Moreover in 2014/15 early winter, the frosts that typically can reduce the survival of the pupae in the soil did not take place. All these conditions allowed the olive fly to be able to fulfil a greater number of generations than usual, locally up to 3–4 generations with the result that many olives were destroyed by the insect (Consorzio Lamma Rete Toscana, 2016). It is widely known that agronomic (relative humidity of summer months, rainfall of whole year) and technological (oil extraction and storage) conditions heavily affect olive oil characteristics. This occurs for both triacilglycerols (TAGs) and minor components (such as polyphenols) content, although these variations are strictly cultivar dependent (Inglese et al., 2011). In this regard, Romero et al. (2003) reported that modifications of lipid biosynthesis (in particular the degree of lipids unsaturation) are positively connected with the rainfall regime in the summer period. Some cultivars in “warm” years (or cultivated in warmer areas) can produce oil with a high linolenic acid content (Peranzana, Pignola, Maurino, Nolca, Cellina di Nardò, Cassanese, Ogliastro), (Lombardo et al., 2008) which may exceed the limits allowed by EU regulations (EUR-Lex, 2016). Furthermore, the oils from several cultivars showed that a decrease in oleic acid results mainly compensated by an increase in saturated and polyunsaturated fatty acids (Lombardo et al., 2008). A relationship between climate and fatty acid composition has been studied for Leccino and Casaliva cultivars, showing that Leccino seemed to be insensitive to seasonal thermal trend during maturation (Tura et al., 2008). These results do not often agree with the so-called Ivanov rule, i.e., “the amount of linoleic acid rises when the temperature decreases, contrary to oleic acid” (Ivanov, 1927; Ivanov, 1929). To estimate these effects on Apulia extra virgin olive oils (EVOOs), more data is needed on the specific behaviour of Apulia region cultivars with respect to these microclimatic and pedoclimatic factors (Palese et al., 2010). In particular, the definition of the characteristic and evaluation of the harvest year effects on Coratina is very important. This latter is the most popular olive cultivar of the Apulia region, accounting for almost 40% of the total country production. For this reason, Coratina together with three popular local cultivars used as “sweeteners” in Coratina-based blends (Del Coco et al., 2014; Girelli, Del Coco & Fanizzi, 2015) (Ogliarola, Cima di Mola and Peranzana) from the Bari and Foggia provinces (Southern Italy, Apulia region) were studied in two different harvesting years (2013/14 and 2014/15, H13 and H14). The aim of a specific project PON “R&C” 2007–2013 (PON01_01958 PIVOLIO) was to analyse the production year effect on monovarietal oils characteristics in order to evaluate also possible geographical classification capabilities. Therefore, a specific Apulian 1H-NMR spectral database has been constructed using 900 EVOO samples obtained from 900 cultivar certified single trees. Finally, the harvest year effect at cultivar and somehow even at the plant level has been studied.

Table 1 List of samples (450/year) and relative areas of origin of samples.

No. samples	Cultivar	Area of origin (subareas)	
480	Coratina	Andria, Barletta, Trinitapoli, Canosa, Minervino murge, Spinazzola, Corato, Bisceglie, Ruvo, Mariotto, Palombaio, Grumo, Toritto, Bitonto, Palo del Colle, Terlizzi, Acquaviva, Cassano	
180	Cima di Mola	Monopoli, Polignano, Alberobello, Putignano, Castellana, Turi, Noci, Conversano, Mola	
120	Ogliarola	Bitonto, Molfetta, Terlizzi	
120	Peranzana	Torremaggiore, San Severo	

Materials and Methods

Sampling

A total of 900 extra virgin olive oils were obtained from four Apulian cultivars (Coratina, Cima di Mola, Ogliarola, Peranzana) in the provinces of Bari and Foggia (Apulia, Italy), collected in two subsequent harvesting periods: 2013/14 (H13) and 2014/15 (H14) (450 samples/year) (Table 1). The cultivars were initially recorded for each tree as assessed by each producer, and successively genetically confirmed for all samples by extraction of genomic DNA from leaves and detection of specific microsatellite markers, according to a standardized procedure (Salimonti et al., 2013). Each olive oil sample was produced from the drupes of a single tree (Piccinonna et al., 2016). Olive harvest were performed from each marked tree (with identification code) at optimal olive ripening stage in different periods, depending on cultivar and growing conditions (dry or well-watered field). Starting from Ogliarola cultivar, which has a natural early olive fruit ripening, the olive harvest operations were activated in the first decade of December and conducted up to the end of January for both the two harvest years (2013/14 and 2014/15). About 25–30 kg of olives per tree were collected by the use of suitable net catchers under the tree canopy, with the help of both mechanical and manual pickers and successively stored in airy boxes, marked with the identification code of the tree. Oil micro-extraction was normally performed within 24 h after the olive harvest. EVOO samples were obtained using the Spremoliva C30 milling machine (Toscana Enologica Mori, Tavarnelle Val di Pesa (FI), Italy), and successively stored in sealed dark glass bottles at room temperature in the dark prior to analysis. As much as 25–30 kg of olives were processed for each working cycle of approximately 2–3 h. The machine was cleaned carefully after each cycle.

NMR measurements

NMR samples were prepared dissolving ∼140 mg of olive oil in CDCl3 and adjusting the mass ratio of olive oil:CDCl3 to 13.5%:86.5%. Next, 600 µL of the prepared mixture were transferred into a 5-mm NMR tube. This ratio was chosen to give the best tradeoff for sensitivity/solution viscosity in spectral acquisition (Bruker Italia, standardized procedure for olive oil analysis). 1H NMR spectra were recorded on a Bruker Avance spectrometer (Bruker, Karlsruhe, Germany), operating at 400.13 MHz, T = 300 K, equipped with a PABBI 5-mm inverse detection probe incorporating a z axis gradient coil. NMR experiments were performed under full automation for the entire process after loading individual samples on a Bruker Automatic Sample Changer (BACS-60), interfaced with the software IconNMR (Bruker). Automated tuning and matching, locking and shimming, and calibration of the 90°  hard pulse P(90°) were done for each sample using standard Bruker routines, ATMA, LOCK, TOPSHIM and PULSECAL, to optimize NMR conditions. For each sample, after a 5-min waiting period for temperature equilibration, a standard one-dimensional (1H ZG) NMR experiments was performed. The relaxation delay (RD) and acquisition time (AQ) were set to 4 s and ∼3.98 s, respectively, resulting in a total recycle time of ∼7.98 s. FIDs were collected into time domain (TD) = 65,536 (64 k) complex data points by setting: spectral width (SW) = 20.5524 ppm (8223.685 Hz), receiver gain (RG) = 4, number of scans (NS) = 16. Accumulation of 16 scans (or even fewer) are usually used for samples where metabolites are present in high concentrations, as in the case of olive oil (Barison et al., 2010; Del Coco et al., 2016).

1H NMR spectra pre-processing and Statistical analysis

The NMR raw data set was pre-processed using Topspin 2.1 and AMIX 3.9.15 (Bruker BioSpin GmbH, Rheinstetten, Germany). The FIDs were multiplied by an exponential line broadening function (0.3 Hz) before Fourier transformation and automatically phased. Spectra were referenced to the TMS signal at 0.00 ppm, used as an internal standard, obtaining good peak alignment. NMR spectra were processed using Topspin 2.1 (Bruker) and visually inspected using Amix 3.9.15 (Bruker, Biospin). Furthermore, spectra were segmented in rectangular buckets of fixed 0.04 ppm width and integrated, using the Bruker Amix software. Bucketing was performed within 10.00–0.5 ppm region, excluding the signal of the residual non-deuterated chloroform and its carbon satellites (7.6–6.9 ppm); total sum normalization was applied to minimize small differences due to total olive oil concentration and/or acquisition conditions among samples. The Pareto scaling method, which is performed by dividing the mean-centered data by the square root of the standard deviation, was then applied to the variables (Gallo et al., 2014; Sundekilde, Larsen & Bertram, 2013). The data table generated by all aligned buckets row reduced spectra was used for multivariate data analysis. Each bucket row represents the entire NMR spectrum, and all the molecules present in the sample. Each bucket in a buckets row reduced spectrum is labeled with the value of the central chemical shift for its specific 0.04 ppm width. The variables used as descriptors for each sample in chemometric analyses are the buckets. Multivariate analyses (MVA) and graphics were obtained using Simca-P version 14 (Umetrics, Sweden) using different procedures: PCA, PLS-DA and OPLS-DA (Lindon, Nicholson & Holmes, 2011). Principal Components Analysis (PCA), an unsupervised pattern recognition method, was performed to examine the intrinsic variation in the data set. To maximize the separation between sample classes, Partial Least-Squares Discriminant Analysis (PLS-DA), was applied. The PLS-DA is the regression extension of PCA, which gives the maximum covariance between the measured data (X variable, matrix of buckets related to metabolites in NMR spectra) and the response variable (Y variable, matrix of data related to the class membership). Beside PLS-DA, also Orthogonal Partial Least-Squares Discriminant Analysis (OPLS-DA) has been applied in MVA. As shown in several metabolomics recent studies, OPLS-DA represent the most recently used technique for the discrimination of samples with different characteristics (such as cultivars and/or geographical origin). OPLS-DA is a modification of the usual PLS-DA method which filters out variation that is not directly related to the response. The further improvements made by the OPLS-DA in MVA resides in the ability to separate the portion of the variance useful for predictive purposes from the not predictive variance (which is made orthogonal). Furthermore, OPLS-DA focuses the predictive information in one component, facilitating the interpretation of spectral data. On other hand, when a four categories (the cultivars) model was used for further classification purposes PLS-DA rather than OPLS-DA was preferred (Boccard & Rutledge, 2013). Both for PLS-DA and OPLS-DA, the quality of the models obtained was assessed by R2 and Q2 values. The first (R2) is a cross validation parameter defined as the portion of data variance explained by the models and indicates goodness of fit. The second (Q2) represents the portion of variance in the data predictable by the model. This latter indicates the model predictive ability, which is extracted according to the internal 7 fold cross-validation method of SIMCA-P software (Holmes et al., 2008; Trygg & Wold, 2002). The minimal number of components required can be easily defined since R2 (cum) and Q2 (cum) parameters display completely diverging behaviour as the model complexity increases. The addition of further unnecessary components to the model can, therefore, easily be detected and avoided.

Cluster difference and quality metrics

Principal Component Analysis (PCA) finds the principal components of data, provides a general overview and underlies the structure of the data. However, by visual inspection of PCA scoreplots only a qualitative general separation of the cluster can be evaluated. In order to quantify the magnitude of the separation of the clusters in the PCA scoreplots, the Mahalanobis distances were calculated (Mahalanobis, 1936). Mahalanobis distances calculated between groups in PCA scores space will closely approximate those calculated on the original data while avoiding possible collinearity of the original variables (Anderson et al., 2008). Mahalanobis distances account for different variances in each direction (PC1, PC2, PC3) and are scale-invariant. Moreover, the quality of the clusters in the number of selected PCs was estimated by using the J2 criterion. This parameter provides a measure of the compactness and identity of the cluster (Worley, Halouska & Powers, 2013).

The J2 criterion is defined as: J2=Sw+SbSw;

with Sw being the within-class scatter and Sb the between-class scatter. A high J2 value means well separated and tight clusters. The cluster distances and the cluster quality criterion J2 were calculated with the open source (GNU General Public License 3.0). PCA-utils (Worley, Halouska & Powers, 2013) software, freely available on https://github.com/geekysuavo/pca-utils, and compiled on a Windows 10 64 bit notebook with the mingw-w64 gcc compiler (http://mingw-w64.org/doku.php) in the msys2 environment (http://msys2.github.io/). The data feed to the software were exported from the PCA analyses performed on Umetrics Simca v. 14 (Umetrics Software, Sweden).

Pairwise Mahalanobis distances

Data exported in table format from Umetrics SIMCA software were successively analyzed with the R statistical environment, Version 3.2.4, on a 64 bit Windows machine (R Development Core Team, 2008), using the RStudio environment, version 0.99.893 (RStudio Team, 2015). Pairwise Mahalanobis distances were calculated with the biotools package (Anderson et al., 2008). Plots were prepared with the ggplot2 library (Wickham, 2009). Some of the data preparation operations were performed with the dplyr package (Wickham & Francois, 2015).

Chemicals

All chemical reagents for analyses were of analytical grade. Deuterated chloroform (CDCl3 99.8%-d) containing tetramethylsilane TMS (0.03% v/v) was purchased from Armar Chemicals (Döttingen, Switzerland).

Results and Discussion

Metabolic profiles of the EVOO samples, characterized by 1H NMR spectroscopy, were studied with multivariate analyses (PCA, PLS-DA, OPLS-DA) performed on bucket reduced 1H NMR spectra (see ‘Materials and Methods’). The original dataset obtained from the spectrum of each sample (221 buckets from the spectral region 10.00–0.50 ppm) was rearranged in a new multivariate coordinate space in which the reduced dimensions (usually 2 or 3 in a model scoreplot) are ordered by decreasing explained variance of the considered data. We first analyzed the olive cultivar influence on the whole dataset. Therefore the harvesting year effect was evaluated by comparing the four cultivar dataset and each of the studied cultivars in the two different campaigns (representative spectra for the four cultivars are shown in Fig. S2).

Figure 1 t[1]/t[2] PCA scoreplot for monovarietal EVOO samples (two components give R2 = 0.72, Q2 = 0.65).

Influence of olive cultivar

As a first attempt, in order to reveal a general data grouping of all the samples, an unsupervised PCA analysis was applied to the whole data (1H NMR-bucket-reduced spectra), revealing the presence of some outliers (22 out of 900 samples, divided over the two years), which have been excluded from the analyses. In the PCA analysis two components explained 72.6% of total variance (44.1%, 28.5% for t[1] and t[2] respectively) describing the samples distribution in the space. Visual inspection of t[1]/t[2] PCA scoreplot, reported in Fig. 1, showed a certain degree of separation in particular for the Coratina samples, that were found essentially at negative values of t[1] component and along t[2] component [0.25–0.26]. A relevant degree of overlap was observed among the three remaining classes, Cima Di Mola, Ogliarola and Peranzana. The exclusion of the Coratina class from the PCA model allowed further separation among the three remaining cultivars, especially for Cima di Mola and Ogliarola (see Fig. S3).

In order to improve the separation among the classes for all the four studied cultivars, supervised PLS-DA and OPLS-DA analyses were performed. In these methods, the identity of each sample group is specified in the model such that the maximum variance of the groups can be attained in the hyperspace. Two performance indicators were used to assess the supervised model complexity and eventual over fitting degree: the cross validation (CV) and the response permutation test (n = 400). The PLS-DA model resulted in six components with R2X = 0.88, R2Y = 0.625 and Q2 = 0.61. The 2D t[1]/t[3] and 3D t[1]/t[2]/t[3] PLS-DA scoreplots showed, in this case, a clear separation for Coratina and Peranzana, and a partial overlapping for Cima di Mola and Ogliarola groups (Figs. 2A and 2C). By examining the loadings (Fig. 2B) of the original variables it was possible to define the molecular components distinctive for each class (cultivar). The Coratina group showed high values of monounsaturated fatty acids (i.e., oleic acid), as shown by corresponding loadings at δH 5.34, 2.02, 1.30, 1.98; the Peranzana class was characterized by high values of polyunsaturated fatty acids (PUFA), as indicated by the loadings at δH 2.74, 2.78, signals of linoleic and linolenic bis-allylic groups respectively. High relative content of saturated fatty acids (δH 1.26, corresponding to the methylene of the saturated acyl group) was found for the Cima di Mola and Ogliarola classes, partially overlapped in the t[1]/t[3] scoreplot.

Figure 2 (A) t[1]/t[3] PLS-DA scoreplot for monovarietal EVOO samples (six components give R2X = 0.88, R2Y = 0.625 and Q2 = 0.61); (B) Loadings plot for the model; the variables indicated ppm in the 1H NMR spectra; (C) three-dimensional t[1]/t[2]/t[3] PLS-DA scoreplot for monovarietal EVOO samples showed a clear separation among samples.

Influence of harvest year

In order to obtain further information on the behaviour of the four cultivars within the studied timespan, unsupervised PCA (see Figs. S4 and S5) and supervised PLS-DA analysis was performed considering the two harvesting years separately.

The first PLS-DA model, obtained from 1H NMR spectra of 2013/14 harvesting year EVOO samples, (six components with R2X = 0.88, R2Y = 0.625 and Q2 = 0.61) showed a samples distribution in the t[1]/t[3] scoreplot (Fig. 3A) which is analogue to that observed in the PLS-DA scoreplot when considering all the samples (from both the two studied harvesting years) (see Fig. 2A). A clear separation of the Coratina group from the remaining classes was still observed especially along the first PLS-DA component (t[1]) (Fig. 3A). A certain degree of separation, in particular on the third component (t[3]), was also observed for Peranzana group, while Ogliarola and Cima di Mola appear considerably overlapped in the scoreplot. Analogously for the PLS-DA t[1]/t[3] scoreplot of the 2014/15 harvesting year, Coratina EVOO samples clearly separated from the other three cultivars (Fig. 3B). In this case, the three remaining classes are characterized by a higher degree of separation, along t[3] component (with respect to the 2013/14 samples). This result suggests that, for the studied cultivars, a potential harvesting year effect seems to influence essentially the metabolic profiles of EVOOs other than Coratina.

Figure 3 t[1]/t[3] PLS-DA scoreplots for monovarietal EVOO samples from (A) 2013/14 harvesting year (five components give R2X = 0.89, R2Y = 0.69 and Q2 = 0.68) and (B) 2014/15 harvesting year (five components give R2X = 0.88, R2Y = 0.66 and Q2 = 0.65).

In order to evaluate the potential effect of the harvesting year on the EVOOs characteristics, we compared the two harvesting years for each cultivar performing unsupervised and supervised analysis on the 1H NMR reduced spectra. The PCA analyses of the two years samples for each cultivar are reported in Figs. 4A, 5A, 6A and 7A. The four cultivars were then analyzed by OPLS-DA in order to increase the differences observed in the PCA analysis (Figs. 4C, 5C, 6C and 7C). In the case of the Coratina class, the unsupervised PCA analysis (Fig. 4A) did not reveal differences between the two periods. A poor separation between the two years could be also observed by performing a supervised OPLS-DA analysis, giving a model in which one predictive and six orthogonal components gave R2X = 0.90, R2Y = 0.85 and Q2 = 0.82 (Fig. 4C). The predictive component explained 5.4% of the total variance and the uncorrelated (orthogonal) components to[1], to[2], to[3], to[4], to[5] and to[6] corresponded to 34.8%, 7.92%, 18.1%, 11.6%, 10.8% and 2.31% of the explained variance, respectively. Interestingly, by examining the loadings (Fig. 4D) of the original variables a higher relative content of saturated fatty acids (δH 1.26, corresponding to the methylene of the saturated acyl group) was observed for the 2013/14 Coratina samples.

Figure 4 (A) PCA t[1]/t[2] scoreplot for Coratina samples (five components give R2 = 0.86 and Q2 = 0.76); (B) Loadings plot for the PCA model; (C) t[1]/to[1] OPLS-DA scoreplot for Coratina samples (1 + 6 + 0 components give R2X = 0.90, R2Y = 0.85 and Q2 = 0.82); (D) S-line plot for the OPLS-DA model; the variables indicate ppm in the 1H NMR spectra.

Figure 5 (A) t[1]/t[2] PCA scoreplot for Cima di Mola samples (four components give R2 = 0.89 and Q2 = 0.83). (B) loadings plot for the PCA model; (C) t[1]/to[1] OPLS-DA scoreplot for Cima di Mola samples (1 + 5 + 0 components give R2X = 0.92, R2Y = 0.95 and Q2 = 0.93). (D) S-line plot for the OPLS-DA model; the variables indicate ppm in the 1H NMR spectra.

Figure 6 (A) t[1]/t[2] PCA scoreplot for Ogliarola samples (five components give R2 = 0.90 and Q2 = 0.80). (B) loadings plot for the PCA model; (C) t[1]/to[1] OPLS-DA scoreplot for Ogliarola samples (1 + 6 + 0 components give R2X = 0.91, R2Y = 0.97 and Q2 = 0.95). (D) S-line plot for the OPLS-DA model; the variables indicate ppm in the 1H NMR spectra.

Figure 7 (A) t[1]/t[2] PCA scoreplot for Peranzana samples (five components give R2 = 0.88 and Q2 = 0.76). (B) loadings plot for the PCA model; (C) t[1]/to[1] OPLS-DA scoreplot for Peranzana samples (1 + 6 + 0 components give R2X = 0.89, R2Y = 0.94 and Q2 = 0.91). (D) S-line plot for the OPLS-DA model; the variables indicate ppm in the 1H NMR spectra.

In the case of the Cima di Mola samples, the unsupervised PCA analysis revealed a certain degree of separation (Fig. 5A) between the two harvest years. This separation was also improved by the supervised OPLS-DA analysis, that gave a good model with one predictive and five orthogonal components (with R2X = 0.92, R2Y = 0.95 and Q2 = 0.93) (Fig. 5C). The predictive component explained 51.1% of the total variance and the uncorrelated (orthogonal) and components to[1], to[2], to[3], to[4], to[5] and to[6] corresponded to 11.7%, 13.3%, 9.8%, 11.6%, 3.8% and 2.2% of the explained variance, respectively. A nice partition of the samples, coming from the two different harvesting years, was clearly observed. A high content of saturated fatty acids (loadings at δH 1.26, corresponding to the methylene of the saturated acyl group) in the samples of the 2013/14 harvesting period was shown from the analysis of the loadings of the original variables, determining the separation between the two groups (Fig. 5D).

A clear partition of the samples was also observed in the case of Ogliarola class. As for the Ogliarola class, the visual inspection of t[1]/t[2] PCA scoreplot highlighted the separation among the two groups of samples (Fig. 6A). Also in this case, the OPLS-DA analysis gave a good model, (one predictive and six orthogonal components, R2X = 0.91, R2Y = 0.97 and Q2 =0.95) (Fig. 6C), improving the separation between the classes. The predictive component explained 21.5% of the total variance and the uncorrelated (orthogonal) and components to[1], to[2], to[3], to[4], to[5] and to[6] corresponded to 27.4%, 19.7%, 15.7%, 3.3%, 1.5% and 1.6% of the explained variance, respectively. Interestingly, a reversal trend for the content of saturated fatty acids was observed for the Ogliarola with respect to Coratina and Cima di Mola samples, showing in this case a relatively higher content of saturated fatty acids (δH 1.26, corresponding to the methylene of the saturated acyl group) for the 2014/15 with respect to 2013/14 samples. Moreover, a high content of linolenic acid (δH 0.9, 2.06, 2.78, 5.38) in the samples of the 2013/14 harvest period was observed from the loadings of the original variables.

In the case of the Peranzana group, the unsupervised PCA analysis revealed a separation between the two harvest periods (Fig. 7A) and this separation was highlighted by the OPLS-DA analysis (Fig. 7C). The good OPLS-DA model resulted in 1 predictive and six orthogonal components, that give R2X = 0.89, R2Y = 0.94 and Q2 = 0.91. The predictive component explained 24.8% of the total variance and the uncorrelated (orthogonal) components to[1], to[2], to[3], to[4], to[5] and to[6] corresponded to 31.6%, 15.4% , 6.6%, 6%, 2.8% and 2.6% of the explained variance, respectively. The separation between the two harvesting periods was due, as also observed for Ogliarola, to the higher content of linolenic acid (δH 0.9, 2.06, 2.78, 5.38) in the 2013/14 samples. On the other hand, similarly to Coratina and Cima di Mola samples, a higher content of saturated fatty acids (δH 1.26, corresponding to the methylene of the saturated acyl group) was observed by examining the S-line loading plot for the 2014/15 with respect to 2013/14 samples (Fig. 7D).

Fatty acid profile in EVOOs is primarily determined by its cultivar, although environmental factors can influence oil quality. Climatic conditions, seasonal weather fluctuations, such as rainfall and temperature can affect the physiological behavior of olive tree and the metabolic profile of its oil (Lombardo et al., 2008; Romero et al., 2003). However the effect of the seasonal variability are strongly cultivar dependent as observed in previous investigations (Inglese et al., 2011; Romero et al., 2003). We observed that Ogliarola samples showed a higher relative content of linolenic acid in 2013/14 (warm year) with respect to 2014/15 harvest year, the last characterized by dramatically intense rainfall and cool temperatures (Consorzio Lamma Rete Toscana, 2016). At the same time, Peranzana group was observed to exhibit a similar high relative content of linolenic acid. In the case of the Cima di Mola samples, a strong content of saturated fatty acids was observed in the 2013/14 harvest year. Harvest year and in particular climatic conditions resulted a discriminating parameter although Coratina EVOOs appear to be the less affected and more stable. From this point of view, the Coratina-based EVOO appears to provide well-defined chemical and sensory characteristics of a genetic and territorial origin (soil and climate) (Fanizzi et al., 2015). The lower harvesting year effect which affects the Coratina with respect to the other cultivars considered in the present study is also evident when considering the Q2 predictivity parameter for the OPLS-DA models of Figs. 4B, 5B, 6B and 7B. Indeed, due to the partial overlap of 2013/2014 and 2014/2015 samples, the prediction ability for the year classification related to Coratina (Q2 = 0.82) is about 10% lower with respect to Ogliarola (Q2 = 0.95), Cima di Mola (Q2 = 0.93) and Peranzana (Q2 = 0.91) cultivars. The results obtained with this semi-quantitative assessment on the influence exerted by the harvesting year on the studied cultivars was been further examined by two additional metrics, namely the Mahalanobis distances and the J2 criterion, calculated for the four cultivars using the data from the PCA analyses shown in Figs. 1, 4A, 5A, 6A and 7A.

Cluster difference and quality metrics: Mahalanobis distances and J2 criterion

The graphical representation of the unsupervised PCA, while giving a good relative idea of group dispersion and distances, cannot be used to directly compare among independently conducted analyses. Moreover, when the amount of cumulative variance needed to show the difference in groups requires more than three PCs, an intuitive graphical representation is not possible. Therefore, we calculated different metrics on clusters in order to obtain quantitative values useful for summarizing our conclusions. The first metric we calculated to give a numerical reference value for the PCAs was the Mahalanobis distance (MAH) among the groups. To the best of our knowledge, few chemometric studies discussed the most appropriate method on estimating the suitable number of components (PCs) to measure this type of distance (Brereton & Lloyd, 2016). For this reason, the Mahalanobis distances were measured according to two different principles: either by keeping the explained variance constant or by selecting a fixed number of components. First of all, Mahalanobis distances were calculated between the two years for the same cultivar in the PCA including all the cultivars (Table 2 with reference to the PCA of Fig. 1), and then between the two years for each single cultivar (Table 3 with reference to the PCAs of Figs. 4A, 5A, 6A and 7A).

Table 2 Mahalanobis distances among the clusters of cultivar calculated in the PCA including all the cultivars (Fig. 1) in the two harvesting years.

To obtain 99% cumulative variance in the PCA containing all the cultivars, 20 principal components (PCs) were used.

	Mahalanobis distances (MAH)	
Cultivar	99% variance (20PCs)	2 PCs (72.6% variance)	
Cima di Mola	8.69	3.87	
Coratina	3.75	0.18	
Ogliarola	11.06	1.89	
Peranzana	7.03	1.47	

Table 3 Mahalanobis distances among the clusters of cultivar calculated for each PCA model including a single cultivar (with reference to the PCAs of Figs. 4A, 5A, 6A and 7A) in the two harvesting years.

To obtain 99% cumulative variance for each calculated for each PCA model including a single cultivar, a different number of PCs was used.

	Mahalanobis distances (MAH)	
Cultivar	99% variance (n. PCs)	2PCs (% variance)	
Cima di Mola	7.63 (16)	4.09 (77.2)	
Coratina	4.44 (21)	0.57 (58.9)	
Ogliarola	10.43 (18)	1.85 (64.9)	
Peranzana	6.93 (19)	2.31 (70.7)	

A general overview can be obtained from the MAH distances calculated for all the PCA models studied and reported in Tables 2 and 3. In particular, when the PCA containing all the cultivars (99% cumulative variance and 20 PCs) was used, the Coratina and Ogliarola cultivars showed the relatively smallest and greatest variation, respectively, for the EVOO obtained in the two successive harvesting campaigns (Table 2, column 1). The same trend and comparable MAH values were obtained in the case of the single cultivar PCA models where, in order to obtain the 99% explained variance, a different number of components for each cultivar was considered (Table 3, column 1). Interestingly, the MAH distances calculated within the two harvesting years and with a single general PCA (Table 2, column 1) or four different cultivar specific PCAs (Table 3, column 1) all fall in the same order of magnitude, with the highest observed value (Ogliarola) less three times that of the lowest (Coratina). It should be noted that the distances calculated only with the first 2 PCs, both for a single general PCA (Table 2, column 2) or four different cultivar specific PCAs (Table 3, column 2) could be misleading, since the amount of variance explained by the first 2 PCs is not close, therefore making the distances not comparable. In fact, Mahalanobis distances calculated for the single PCA first two PCs (Table 2, column 2) showed Ogliarola and Peranzana having a comparable behaviour, with Cima di Mola appearing to be the most sensitive cultivar to the harvesting year related oil variation. Cima di Mola appeared to be the most sensitive cultivar to the harvesting year effect also in the case of the four different, cultivar specific, PCAs (Table 3, column 2), while Ogliarola and Peranzana showed a similar MAH. Nevertheless, even using only the first 2 PCs both for a single general PCA (Table 2, column 2) or four different cultivar specific PCAs (Table 3, column 2) the calculated MAH distances clearly show that the Coratina EVOO could be considered the less sensitive to the harvesting year related variations.

Finally, in order to measure the compactness and identity of each cluster we applied the J2 criterion. The J2 values were calculated for all the cultivars for both the two harvesting years, in the single PCA model built for the whole dataset. The J2 values for each cultivar could describe also the cultivar specific internal variability in each harvesting year. A high value of J2 indicates well separated and tightly clustered groups of samples. Although J2 values could be also calculated for 99% variance (20 PCs) (see Table S1), for clarity reasons we have compared in Table 4 the obtained J2 values in the cases of two or three considered PCs. In the first case, the J2 values reported the scatter of each cluster in first two dimensions, for both the harvesting years, describing shape and relative positions for the ellipses containing each group in the bidimensional plane. Similarly, the J2 values reported for 3 PCs described the scatter of each cluster in the first three dimensions, defining shape and relative positions for the ellipsoids containing each group in the Cartesian space. Differently from the previous metric, the J2 values maintain a consistent behaviour, as represented in the examined cases (Table 4). The cluster integrity and separation are very similar both in the two cases (2 and 3 PCs) and for 99% variances, where the J2 criterion is able to give a measure of the quality of clustering also in a n-dimensional space (20 PCs). Cima di Mola showed the highest J2 values for both the two harvests, indicating well-separated compact clusters, especially for 2014/15. Ogliarola and Peranzana showed similar behaviors for both the two harvest years, with a very high J2 value (108.67) obtained for Peranzana in the 2014/15 harvest year. Finally, low J2 values reported for Coratina indicated a very high homogeneity of EVOO samples for both the two harvests and number of considered PCs. This data confirmed the Coratina cultivar to be the less sensitive to the harvesting year related variations, as already discussed for the MAH distances.

Table 4 J2 criterion calculated for each class (cultivar) in the two harvesting years by considering the first two, three, and four PC components in the PCA performed on the whole dataset (reported in Fig. 1).

	72.6% variance (2 PCs)	80.2% variance (3 PCs)	85.3% variance (4 PCs)	
J2 values	2013/14	2014/15	2013/14	2014/15	2013/14	2014/15	
Cima di Mola	24.92	46.33	113.13	321.78	119.309479	924.092661	
Coratina	4.01	6.51	4.63	10.60	10.82181	10.932898	
Ogliarola	15.68	20.29	13.89	72.43	40.004852	1584.137069	
Peranzana	8.00	16.36	13.20	108.67	26.428358	682.562196	

The absolute values increased in the case of higher dimensionality of the data, but the trends are maintained. Except for the Coratina class, the J2 values are consistently lower for the 2013/14 with respect to 2014/15 harvest year, indicating both higher overlap and diffusion of the clusters. In Fig. 1 and Fig. S4 the J2 values can be visually interpreted: Coratina has the widest variance (apparently due to the presence of two possible subgroups), while the Cima di Mola has the tightest cluster. Peranzana has few overlaps with the other cultivars, but at the same time has a high variance. The same results are visible qualitatively for the 2014/15 harvest, but all the clusters appear to be more spaced among themselves (Fig. S5). When the number of PCs examined goes to four, the general trends are maintained, with the spread for the cultivars increasing, except for the Coratina.

Figure 8 Schematic representation of pairwise (A) and random-wise (B) structure of the considered data in the harvest years comparison.

Pairwise Mahalanobis distances

As largely described in literature (Jonsson et al., 2015; Westerhuis et al., 2010), NMR-based metabolomics is a very powerful techniques but affected by different types of bias, for example when a large intrinsic variation is present between the subjects or data acquired over long periods of time were combined and analyzed. This is particularly evident in clinic and diagnostic studies, when intrinsic variation between human and/or biological samples can largely influence metabolome. For this reason, multivariate analysis, and in particular the supervised OPLS-DA technique, works reducing the uncorrelated and/or unwanted variation by applying a separation between the subjects variation from the studied effect, focusing the attention on the effect of interest. In our work, we applied this criterion when considering the whole dataset and the clustering quality of unsupervised models. In the next and last step of this work, a paired (rather than random) structure of the data was considered (Fig. 8), in order to provide information about the difference across the study population.

To the best of our knowledge, there are no other reports on harvesting year effect for EVOO samples originated from individual olive trees with genetically established cultivar identification.

Using the PCA data, the pairwise squared Mahalanobis distances (MADs) were calculated for all the couples of samples (2013/14 and 2014/15 harvest year). The samples having no correspondence in the other harvest (due to harvesting or oil production) were excluded from this analysis. The square roots of said values were calculated and then divided in two sets: a set consisting of the distances between the EVOO obtained in the two harvest years from a single plant, and a set containing the distances of a single sample in the harvest 2013/14 from all the samples of the same cultivar in the following harvest (2014/15). The modulus of the distances was considered. The first distribution can be envisioned as a trace of the difference in the global environment that a single plant experienced in the two harvest year, while the second distribution could be held to represent an all-including representation of the cultivar variability between the two harvests.

Notwithstanding the different sample sizes for the four cultivars, an information that can be extracted by the data is that the couples of cultivars Coratina–Ogliarola and Cima di Mola-Peranzana share a similar behavior. In fact, Coratina and Ogliarola have a smaller mean distance and median between the two harvests (Fig. 9 and Table 5), indicating that on average the samples had a lower variability in the two harvests than Cima di Mola and Peranzana, although Coratina has a much higher number of measured samples, so a higher variability would not have been unexpected. On the other hand, standard deviations are close for all the cultivars except a slightly higher value for Coratina, as expected on the basis of the much wider sample size. Cima di Mola and Peranzana share higher medians and means, with a more symmetric distribution around the mean. These data suggest a higher variability of the EVOO obtained from a single plant in the two years, a possible consequence of significant differences in the climatic sub regions where the plant were located, between two harvest years.

Figure 9 Kernel density plot of the Mahalanobis distances of the EVOOs obtained from a single plant in the two successive harvests.

Summary data for the distributions are described in Table S2.

Table 5 Comparison of the summary data (median, mean, standard deviation, S.D.) for the pairwise distances calculated for the EVOO obtained from a single plant in two successive harvests (plant–plant) and the distances of the EVOO of a single plant from all the EVOOs obtained for the whole cultivar in the following harvest (plant–cultivar).

	Plant–plant	Plant-cultivar	
Cultivar	Median	Mean	S.D.	Median	Mean	S.D.	
Coratina	0.168	0.193	0.039	0.168	0.190	0.081	
Ogliarola	0.169	0.184	0.024	0.179	0.189	0.065	
Peranzana	0.216	0.219	0.028	0.204	0.208	0.052	
Cima di Mola	0.206	0.214	0.029	0.217	0.219	0.062	

The trends in the aggregated data are similar (Fig. 10 and Table 6). The overall shape of the distributions and the average distances remain close, with Peranzana featuring slightly smaller mean and median. This could be attributed to the change in spread of the Peranzana in Harvest 2014/15, with the formation of two subgroups, which can be easily visually identified in Figs. 7A and 7B, confirmed by the doubling of the J2 value of H14 with respect to H13.

Figure 10 Kernel density plot of the cumulative Mahalanobis distances of a single plant EVOOs from all the plants of the cultivar in the following harvest.

Minimum, median, mean, and maximum values are summarized in Table 6.

Table 6 Cultivar minimum, median, mean, and maximum distance for the set of the Mahalanobis distances of every sample of the 2013/14 from every sample of the 2014/15 harvest.

Cultivar	Minimum	Median	Mean	Maximum	
Coratina	0.042	0.168	0.190	0.713	
Ogliarola	0.060	0.179	0.189	0.469	
Peranzana	0.062	0.204	0.208	0.426	
Cima di Mola	0.084	0.217	0.219	0.472	

Conclusions

The collection of 1H NMR data on EVOO prepared exclusively from olives collected from a single and genetically identified plants, allowed us to quantitatively assess the differences among four different cultivars and in two subsequent harvesting years. The two harvests were performed in two years showing exceptionally significant differences in the rainfall volumes and in maximum temperatures, resulting in a very different overall production, for all the considered collecting areas.

Preliminary exploratory PCAs, showed already a certain separation among the four studied cultivars in the two considered harvesting years. All the studied monocultivar oils, could be clearly differentiated by supervised MVA in both harvesting years. Moreover, supervised multivariate analysis showed different behaviour in the two subsequent harvesting years for each considered cultivar. Indeed, the studied monocultivar oils resulted, for each cultivar, as separate groups for the two subsequent harvesting years when the OPLS-DA was performed. The OPLS-DA Q2 predictivity values clearly indicate the Coratina Cultivar as the less affected by the harvesting year effects. Comparison of Q2 predictivity results with the PCA Mahalanobis distances of clusters, and J2 value compactness widely agree on describing the behaviour of the cultivars among themselves and in the different harvest. Pairwise and random-wise consideration of the data, in the harvest years comparison, allowed detection of the consequences for significant differences in the climatic sub regions where the plants were located. These latter resulted minimal in the overall considered study area. All the analysis made on the 1H-NMR based metabolomics concur to confirm that the overall variations for Coratina EVOO are the smallest, therefore making Coratina based blends useful to maintain EVOO characteristics even in the case of seasons with very different climatic conditions.

Supplemental Information

Supplemental Information 1 Supplementary information

Click here for additional data file.

We would like to thank Oliveti Terra di Bari Soc. Coop. Agricola and Olearia Basile Snc for collaboration and support in the present work.

Abbreviations

MAH Mahalanobis distances

PCA Principal Component Analysis

PLS-DA Partial Least-Squares Discriminant Analysis

OPLS-DA Orthogonal Partial Least-Squares Discriminant Analysis

Additional Information and Declarations

Competing Interests

Author Contributions

Data Availability

The authors declare there are no competing interests.

Chiara R. Girelli and Laura Del Coco conceived and designed the experiments, performed the experiments, analyzed the data, wrote the paper, prepared figures and/or tables, reviewed drafts of the paper.

Paride Papadia analyzed the data, wrote the paper, prepared figures and/or tables, reviewed drafts of the paper.

Sandra A. De Pascali performed the experiments.

Francesco P. Fanizzi conceived and designed the experiments, contributed reagents/materials/analysis tools, wrote the paper, prepared figures and/or tables, reviewed drafts of the paper.

The following information was supplied regarding data availability:

The raw data was supplied for review but it is owned by a third-party (PIVOLIO project partners) who have not given their permission to publish alongside this manuscript since it is part of the general PIVOLIO project database, a governed resource hosted at: http://www.pivolio.it/.

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
