# Peer review of "Harvest year effects on Apulian EVOOs evaluated by 1H NMR based metabolomics"

_PeerJ, doi:10.7717/peerj.2740_

## Round 0.1 · original submission · Minor Revisions

Three reviewers appreciated the quality of your study.

Please address their comments and submit a minor revised version as soon as possible.

Reviewer 1 ·

Basic reporting

The manuscript described very well the state of art related to the topics under study.

Experimental design

The experimental design was well organized.

Validity of the findings

The findings had a very high validity.

Additional comments

The authors used very robust multivariate analysis, including in the model a significant number of samples. The results are quite clear, highlighting the effects proposed by the authors in the objectives. In addition, the authors have interpreted and effectively discussed the results obtained.

Reviewer 2 ·

Basic reporting

I think that the paper has been written in a clear, unambiguous English language.
I suggest to check the following rows and to correct the sentences as indicated:
• 36: The first “Still” can be substituted with nevertheless
• 66: Bigger/greater (“than usual”) needs to be inserted
• 278: “and OPLS-DA” needs to be deleted
Considering the introduction, I think it is well written and centred on the focus of the work.
Materials and methods are exhaustive, clear and, even if generally I don’t like when this part of the paper is too much descriptive, considering the paragraphs on the tools that you used in order to estimate the differences between harvesting seasons, I found the explanation that you added on how Mahalanobis distances and the J2 criterion work useful for the readers since it was explained clearly and quickly.
Finally, I carefully read the results and discussion and I liked how you structured the writing.

Experimental design

You employed a simple experimental design that, with the good statistical approach applied, allowed you to get the information that you were searching for.

Validity of the findings

The only criticism is more a general consideration (anyway I strongly suggest you to think about it for the next work) is that from so much overstress the multivariate statistical approach (an absolutely needed tool trying to search for strong variation inside your complex dataset) you could move to different and more specific analysis (such as HPLC detailed quantification of the fatty acids on which you were focusing or, even more interesting from a commercial and practical point of view, a panel test to assess if panellists were able to feel the differences that you found) in order to get more information on your very nice and exhaustive pool of samples.

Reviewer 3 ·

Basic reporting

It is an interesting manuscript with a study of many samples. The objective of to get a relationship between the harvest year and the olive oils of different cultivars is quite interesting too. It is a very nice application of the statistic in the knowledge of the effects of harvest year.
All the figures are relevant, only there is one, in the supplemental information, Figure S2, that should be better to add a legend.
In my opinion a figure with some 1H-NMR spectra, which show the spectra of the oils from different cultivars could improve the paper.
Finally, in the Reference part, there are two of them that the volume and pages are missing.
Line 563 it must be include 30:134-143
And Line 578 insert 118:1380-1388

Experimental design

The method are describe with sufficient information.
Only could be better indicate why the mass ratio of olive oil:CDCl3 in the NMR sample is that, and no other one (also concentrated).

Validity of the findings

The data are robust, and the conclusions are appropriately stated, and are limited to the results that are supported in the manuscript.

Additional comments

Very interesting work.

---

## Round 0.2 · accepted · Accept

I have now read your revised article and I am largely satisfied by the quality of your work and the detailed reply to reviewers.